# A real-world multicenter cross-sectional observational study to assess the clinical profile of peripheral neuropathy in patients with diabetes

**Ashu Rastogi**[1]*, **Venkatesan Ravindranath**[2], **Anupama Dubey**[3], **Dilip Gude**[4], **Manish Agarwal**[5], **Hiren Prajapati**[6], **Willem Jan Verberk**[7]

1 Department of Endocrinology and Metabolism Post Graduate Institute of Medical Education and Research (PGIMER), Chandigarh, India, 2 Prabhu Diabetes Speciality Centre, Trichy, Tamil Nadu, India, 3 Maharishi Diabetes and Foot Care Centre, Indore, Madhya Pradesh, India, 4 Yashoda Hospitals, Somajiguda, Hyderabad, India, 5 Medilink Hospital Research Centre, Ahmedabad, Gujarat, India, 6 Department of Medical Affairs, Eris Lifesciences Ltd., Ahmedabad, Gujarat, India, 7 CARIM School for Cardiovascular Diseases, Maastricht University, Maastricht, The Netherlands

* ashuendo@gmail.com

**Data Availability Statement:** All relevant data are within the paper and its Supporting Information files.

## Abstract

### Background

There are limited studies on the prevalence of diabetic peripheral neuropathy (DPN) and related foot deformities in patients with T2DM from India.

### Aim

To investigate the prevalence, characteristics, and risk factors for foot deformities in Asian-Indian individuals with T2DM and DPN.

### Methods

We analyzed 4290 patients (32.3% female, 67.7% male, mean age 51.1 ± 9.3 years) using a cross-sectional, retrospective observational method, focusing on signs and symptoms of foot complications and neuropathy.

### Results

Dry Skin (44%), infection (19.7%), and ingrown toenails (16.6%) were the foremost prevalent foot health conditions. The most common neuropathic symptoms were burning (35.4%), muscle cramps (31.5%), and loss of sensation (26.6%). Multivariable logistic regression analysis identified nephropathy (OR 3.96 [95% CI: 3.02–5.20]), retinopathy (OR 3.85 [95% CI: 2.72–5.48]), coronary disease (OR 3.48 [95% CI: 2.42–5.04]), COVID-19 history (OR 2.37 [95% CI: 1.73–3.26]), smoking (OR 2.13 [95% CI: 1.56–2.91]), hypertension (OR 2.10 [95% CI: 1.63–2.73]), dyslipidemia (OR 2.09 [95% CI: 1.62–2.69]), alcohol use (OR 1.57 [95% CI: 1.14–2.15]), and high HbA1c (OR 1.29 [95% CI: 1.16–1.42]) as

**Funding:** This study was made possible with help of Eris Lifesciences Ltd. The funders had no role in study design, data collection and analysis, decision to publish, or preparation of the manuscript.

**Competing interests:** The authors have declared that no competing interests exist.

significant predictors ($p < 0.001$) of increased risk for multiple foot health complications. Diabetes duration showed no significant correlation with increased risk for multiple foot complications.

## Conclusion

This study revealed a significant incidence of foot deformities and neuropathic symptoms in Indian T2DM patients, influenced by various lifestyle and medical factors. The lack of correlation between diabetes duration and foot complications in the present study highlights the need for enhanced diabetes management and early detection strategies in India.

## Introduction

The global prevalence of diabetes, including in India, is increasing. Currently, there are 537 million adults with diabetes worldwide, a number expected to rise to 643 million by 2030 and 783 million by 2045. In India, diabetes prevalence increased from 9% in 2011 to 9.6% in 2021 and is projected to reach 10.4% by 2030 [1].

Long-term diabetes often results in various vascular complications. The most prevalent clinical manifestation is damage to the nerves, which occurs in approximately half of all diabetes patients affecting both the peripheral and autonomic nervous system. [2] This condition, referred to as neuropathy, shows up in different forms [3]. Diabetes peripheral neuropathy (DPN) is considered a main risk factor for amputation, and hence a significant cause of morbidity or even mortality in diabetes [4].

In DPN the protective sensation in the extremities, mainly in the feet, deteriorates [5]. This deterioration makes it difficult for individuals to detect foot trauma, increasing the risk of neglecting foot care when the risk of foot ulcers is high. Hyperglycemia further exacerbates this problem by impairing leukocyte and complement function, which increases the risk of invasive infections [6]. After healing, ulcers have a high recurrence rate, estimated at 40% per person per year in Europe [7]. Foot deformities such as calluses, bunions, hammer toes, claw toes, and flat feet can also increase the risk of ulcer development. Therefore, early screening to identify these structural abnormalities, followed by proper management of patients at risk of developing foot ulcers, is a key recommendation in many diabetic foot care guidelines. Early recognition and intervention can reduce the likelihood of lower extremity amputation [8–10].

In DPN, foot deformities like claw toes and hammer toes are more common compared to those without diabetes. The precise cause of these deformities is not completely clear, but it appears to be linked to an imbalance in muscle function. Specifically, this imbalance is due to atrophy of the intrinsic muscles of the foot, which affects the normal balance between muscles that extend and flex the toes. In conditions like claw and hammer toe deformities, there is increased pressure either under the heads of the metatarsal bones or at the tip of the toe. This additional pressure raises the risk of developing ulcers in these specific areas of the foot [11].

Research on the prevalence of neuropathy and foot deformities / foot health complications in diabetic patients has primarily been conducted in Western developed countries, resulting in a notable deficiency of comparable data from developing regions, particularly South Asia [12]. This scarcity of data is especially concerning given India's designation as the global epicenter of diabetes, highlighting the need for more focused research in this area. Our study, therefore, aimed to investigate the prevalence and characteristics of foot deformities and their associated risk factors in Indian outpatients with Type 2 Diabetes Mellitus (T2DM) diagnosed with DPN.

A cross-sectional, retrospective observational method was utilized to evaluate the variety of signs and symptoms in this patient group.

## Material and methods

### Methods

Data collection for the study was conducted between October 8, 2023, and November 20, 2023, with patients coming from primary, secondary, and tertiary care settings across 543 centers in India, covering the following states: Jammu, Kashmir, Punjab, Rajasthan, Delhi, Bihar, Uttar Pradesh, Madhya Pradesh, Odisha, West Bengal, Assam, Andhra Pradesh, Tamil Nadu, Karnataka, Kerala, Maharashtra, and Gujarat. In a single clinical visit, several data points were collected including demographic and anthropometric details like age, gender, height, and weight. Information on habits such as alcohol consumption and smoking, as well as medical history including diabetes duration, glycated hemoglobin (HbA1c) levels, and current treatment, were also registered. Additionally, the occurrence of co-morbid conditions such as hypertension, dyslipidemia, coronary artery disease, diabetic nephropathy, diabetic retinopathy, and any neurological disorders was recorded.

To assess symptoms indicative of DPN, patients were asked about any sensory disturbances they may have experienced, including burning sensations, tingling, muscle cramps, and a reduced sense of touch. The length of time the patient had experienced these symptoms was also documented. Following this, a thorough examination of the patients' foot health was conducted to detect complications like ingrown toenails, toe deformities (hammer, claw and mallet toe), calluses, infections, bunions, dry skin, skin maceration and corns. A specialized medical examination tool, called the Diabetic Foot Scanner (www.eris.co.in/diabetic-care), which consists of a concave mirror to be placed on the floor over which patient feet (plantar aspect) could be observed by the treating physician was utilized to optimize the visual inspection of the foot. (Tool details in S1 File).

Participants gave their written consent prior to the physical examination. The research protocol was approved by the Medilink ethics committee at Medilink Hospital Research Centre, Ahmedabad, Gujarat, India. In addition, the study was conducted in accordance with the Helsinki Declaration.

### Statistical analysis

A retrospective examination was conducted to analyze the association between patient demographic features and co-morbidities with health-related foot conditions. Characteristics of patients were presented using means ± standard deviations (SD) for continuous variables and frequencies (percentages) for categorical variables. For categorical variables, the Chi-square test was applied, while for continuous variables, depending on their distribution, either the Student's t-test or the Wilcoxon rank-sum test was used.

The statistical analysis concentrated on determining the prevalence and exploring the relationships between various predictors and foot-related health conditions. In addition, the number of foot health complications per patient was calculated and incorporated to the statistical analysis to compare patients without any foot health complication to those with three or more foot heath complications. The rationale for selecting a combined score of $\geq 3$ foot health complications was based on an arbitrary definition, which identified this threshold as indicative of significant foot deformities [13].

Separately, prevalence of neuropathic symptoms (burning, muscle cramps, loss of sensation, tingling) and its association with foot related health conditions were calculated. To determine the most significant predictors of each foot-related health condition, we utilized logistic

regression with a backward elimination approach. This method starts with a comprehensive model incorporating all potential predictors. It then progressively removes the least significant predictor, identified by the highest p-value, until only variables that substantially contribute to the model are retained. The outcomes of the final models were presented using Odds Ratios (ORs), 95% Confidence Intervals (CIs), and p-values for each predictor across all health conditions studied.

## Results

### Patients characteristics

A cohort of 4,290 patients (1385 females (32.3%) and 2905 males (67.7%)) with T2DM and diagnosed with neuropathy were included. The average age of the population was 51.1 ± 9.3 years, and the average HbA1c level was 7.8%, with a similar distribution across genders (Table 1). Among the included patients, some were diagnosed with hypertension (n = 2500, 58.3%), dyslipidemia (n = 1453, 33.9%), coronary artery disease (n = 444, 10.3%), diabetic nephropathy (n = 1166, 27.2%), and diabetic retinopathy (n = 553, 12.9%).

Regarding metabolic control, the mean random blood glucose level was significantly higher in males (180.3 mg/dL) compared to females (169.5 mg/dL). Smoking and alcohol intake were markedly higher among male participants (35.4% and 34.9%, respectively) compared to females (2.6% and 2.9%, respectively). The average duration of diabetes was slightly longer in males (4.8 years) than in females (4.4 years). Overall, 914 patients (21.3%) had no foot health complications, while 617 patients (14.4%) experienced three or more such complications. In total, 3157 patients (73.6%) had one or more neuropathic symptoms, and 350 patients (8.2%) had three or more. The prevalence of foot health complications was similar in males and females.

### Prevalence of foot health complications

As shown in Fig 1, Dry Skin Foot (n = 1899, 44% of all patients) emerged as the most prevalent foot health complication, followed by Infection (n = 847, 19.7% of all patients) and Ingrown Toenails (n = 712, 16.6% of all patients). The least prevalent conditions were Toe Deformities (n = 444, 10.3% of all patients) and Calluses (n = 380, 8.9% of all patients).

### Prevalence of neuropathic symptoms

Fig 2 demonstrates that Burning was the most frequent neuropathic foot symptom, reported by 1437 subjects, representing 35.4% of all evaluated patients. Muscle Cramps were experienced by 1282 subjects (31.5%), while Loss of Sensation was noted by 1081 patients (26.2%). Tingling was the least common symptom, affecting 1032 patients (25.3%).

### Logistic regression analysis

Of the 4290 patients analyzed, 914 (21.3%) had no Foot Health Complications, while 617 (14.4%) experienced three or more Foot Health Complications.

Table 2 presents the results of a multilogistic regression analysis: Compared to patients without foot health complications, Diabetes Nephropathy (OR 3.96 [95% CI: 2.72–5.48]) and Diabetic Retinopathy (OR 3.85 [95% CI: 2.72–5.48]) were the strongest predictors for those with three or more complications, followed by CAD (OR 3.48 [95% CI: 2.42–5.04]) and a history of COVID-19 (OR 2.37 [95% CI: 1.73–3.26]).

For individual foot health issues, Dyslipidemia showed a strong association with Bunions (OR 1.67 [95% CI: 1.37–2.02]), and Diabetic Retinopathy was particularly linked to Ingrown

**Table 1. Characteristics of the studied population with diabetes.**

| | Female (N = 1385) | Male (N = 2905) | Total (N = 4290) | p value |
|---|---|---|---|---|
| Age [yrs] | 50.3 (9.4) | 51.5 (9.3) | 51.1 (9.4) | < 0.001 |
| HbA1c [%] | 7.7 (1.2) | 7.8 (1.2) | 7.8 (1.2) | 0.231 |
| Random Blood Glucose [mg/dL] | 169.5 (72.0) | 180.3 (69.1) | 176.8 (70.2) | < 0.001 |
| Ingrown Nails | 231 (16.7%) | 481 (16.6%) | 712 (16.6%) | 0.921 |
| Hammer Toe | 136 (9.8%) | 308 (10.6%) | 444 (10.3%) | 0.431 |
| Callus | 110 (7.9%) | 270 (9.3%) | 380 (8.9%) | 0.145 |
| Foot Infection | 254 (18.3%) | 593 (20.4%) | 847 (19.7%) | 0.111 |
| Bunion | 161 (11.6%) | 356 (12.3%) | 517 (12.1%) | 0.553 |
| Corn | 134 (9.7%) | 336 (11.6%) | 470 (11.0%) | 0.064 |
| Dry Skin Foot | 632 (45.6%) | 1267 (43.6%) | 1899 (44.3%) | 0.214 |
| Maceration | 182 (13.1%) | 425 (14.6%) | 607 (14.1%) | 0.191 |
| Tingling | 373 (26.9%) | 714 (24.6%) | 1087 (25.3%) | 0.098 |
| Burning Sensation | 479 (34.6%) | 1040 (35.8%) | 1519 (35.4%) | 0.436 |
| Loss of Sensation | 359 (25.9%) | 765 (26.3%) | 1124 (26.2%) | 0.773 |
| Muscle Cramps | 415 (30.0%) | 935 (32.2%) | 1350 (31.5%) | 0.143 |
| Smoking Status | 36 (2.6%) | 1029 (35.4%) | 1065 (24.8%) | < 0.001 |
| Alcohol Intake | 40 (2.9%) | 1015 (34.9%) | 1055 (24.6%) | < 0.001 |
| Duration Of Diabetes | 4.4 (3.0) | 4.8 (3.0) | 4.7 (3.0) | < 0.001 |
| Oral Anti-diabetes Durgs (OAD) | 1240 (89.5%) | 2568 (88.4%) | 3808 (88.8%) | 0.273 |
| OAD & Insulin | 145 (10.5%) | 337 (11.6%) | 482 (11.2%) | 0.273 |
| Hypertension | 782 (56.5%) | 1718 (59.1%) | 2500 (58.3%) | 0.096 |
| Dyslipidemia | 442 (31.9%) | 1011 (34.8%) | 1453 (33.9%) | 0.062 |
| Coronary Artery Disease | 142 (10.3%) | 302 (10.4%) | 444 (10.3%) | 0.886 |
| Diabetic Nephropathy | 385 (27.8%) | 781 (26.9%) | 1166 (27.2%) | 0.530 |
| Diabetic Retinopathy | 187 (13.5%) | 366 (12.6%) | 553 (12.9%) | 0.409 |
| Neurological Condition | 159 (11.5%) | 351 (12.1%) | 510 (11.9%) | 0.569 |
| History of COVID-19 Infection | 263 (19.0%) | 508 (17.5%) | 771 (18.0%) | 0.231 |
| Foot Health Complications | 1.3 (1.1) | 1.4 (1.1) | 1.4 (1.1) | 0.092 |
| Neuropathic Symptoms | 1.2 (1.0) | 1.2 (1.0) | 1.2 (1.0) | 0.640 |
| Foot Health Complications | 1075 (77.6%) | 2301 (79.2%) | 3376 (78.7%) | 0.234 |
| Foot Health Complications (quartiles) | | | | 0.440 |
| - 0 | 310 (22.4%) | 604 (20.8%) | 914 (21.3%) | |
| - 1 | 573 (41.4%) | 1180 (40.6%) | 1753 (40.9%) | |
| - 2 | 314 (22.7%) | 692 (23.8%) | 1006 (23.4%) | |
| - ≥3 | 188 (13.6%) | 429 (14.8%) | 617 (14.4%) | |
| Neuropathic Symptoms (quartiles) | | | | 0.154 |
| - 0 | 386 (27.9%) | 747 (25.7%) | 1133 (26.4%) | |
| - 1 | 542 (39.1%) | 1171 (40.3%) | 1713 (39.9%) | |
| - 2 | 333 (24.0%) | 761 (26.2%) | 1094 (25.5%) | |
| - ≥3 | 124 (9.0%) | 226 (7.8%) | 350 (8.2%) | |

Nails (OR 2.00 [95% CI: 1.61–2.48]). The risk of Toe Deformities was significantly elevated in patients with Diabetes Nephropathy (OR 1.51 [95% CI: 1.21–1.87]).

HbA1c demonstrated a modest but significant association with the presence of three or more foot health complications (OR 1.29 [95% CI: 1.16–1.42]). A negative association was observed between the male gender and the presence of Ingrown Nails (OR 0.81 [95% CI: 0.66–

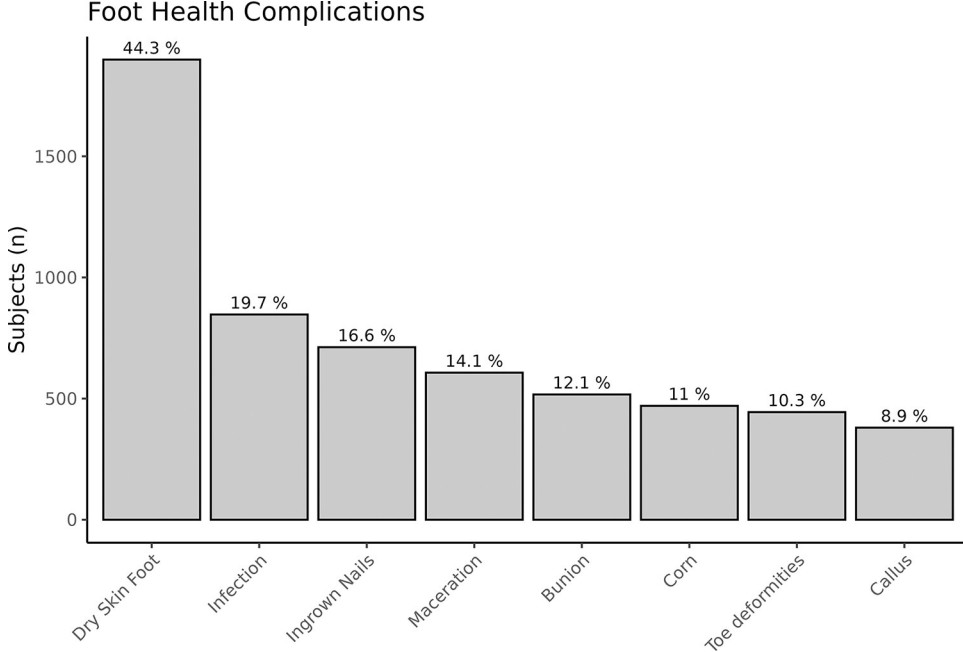

**Fig 1. The bar graph illustrates the percentage of subjects experiencing various foot health complications.**

0.99]). Lastly, Age, Duration of Diabetes, and OAD use showed minimal associations with foot health complications.

Table 3 shows that Callus was significantly associated with all examined neuropathic symptoms in T2DM patients with DNP, particularly with burning sensations (OR 1.78 [95% CI: 1.43–2.21]) and loss of sensation (OR 1.56 [95% CI: 1.23–1.96]). Ingrown Nails also showed a strong relationship across all symptoms, with the highest associations seen with burning sensations (OR 1.90 [95% CI: 1.61–2.24]) and muscle cramps (OR 1.69 [95% CI: 1.42–2.01]). Dry Skin Foot showed a relatively lower association, yet was a significant predictor for loss of sensation (OR 1.69 [95% CI: 1.47–1.95]) and muscle cramps (OR 2.01 [95% CI: 1.76–2.30]). Corn presented a substantial link specifically to muscle cramps (OR 2.14 [95% CI: 1.75–2.62]). In contrast, Infection and Maceration, while associated with several symptoms, did not have such a pronounced impact as the aforementioned conditions.

## Discussion

The present study found that among 4290 T2DM patients with DPN, dry skin was the most common foot health complication, reported by 44% of patients, followed by infections and ingrown toenails. Burning sensations were identified as the predominant neuropathic symptom, affecting 35% of the cohort. Lifestyle factors, such as smoking and alcohol consumption, emerged as significant risk factors for foot complications. The analysis also demonstrated strong associations between systemic health conditions (hypertension, dyslipidemia, coronary artery disease, diabetic retinopathy, and diabetes nephropathy) and an increase in foot health complications. Furthermore, a history of COVID-19 infection was significantly linked to an increased risk of multiple foot health complications. Within this spectrum, an elevated HbA1c level was identified as a significant factor, albeit with a relatively lower impact on foot health issues than the aforementioned factors. Additionally, the study found that specific foot conditions, particularly ingrown nails and calluses, were significantly associated with neuropathic symptoms like burning and muscle cramps.

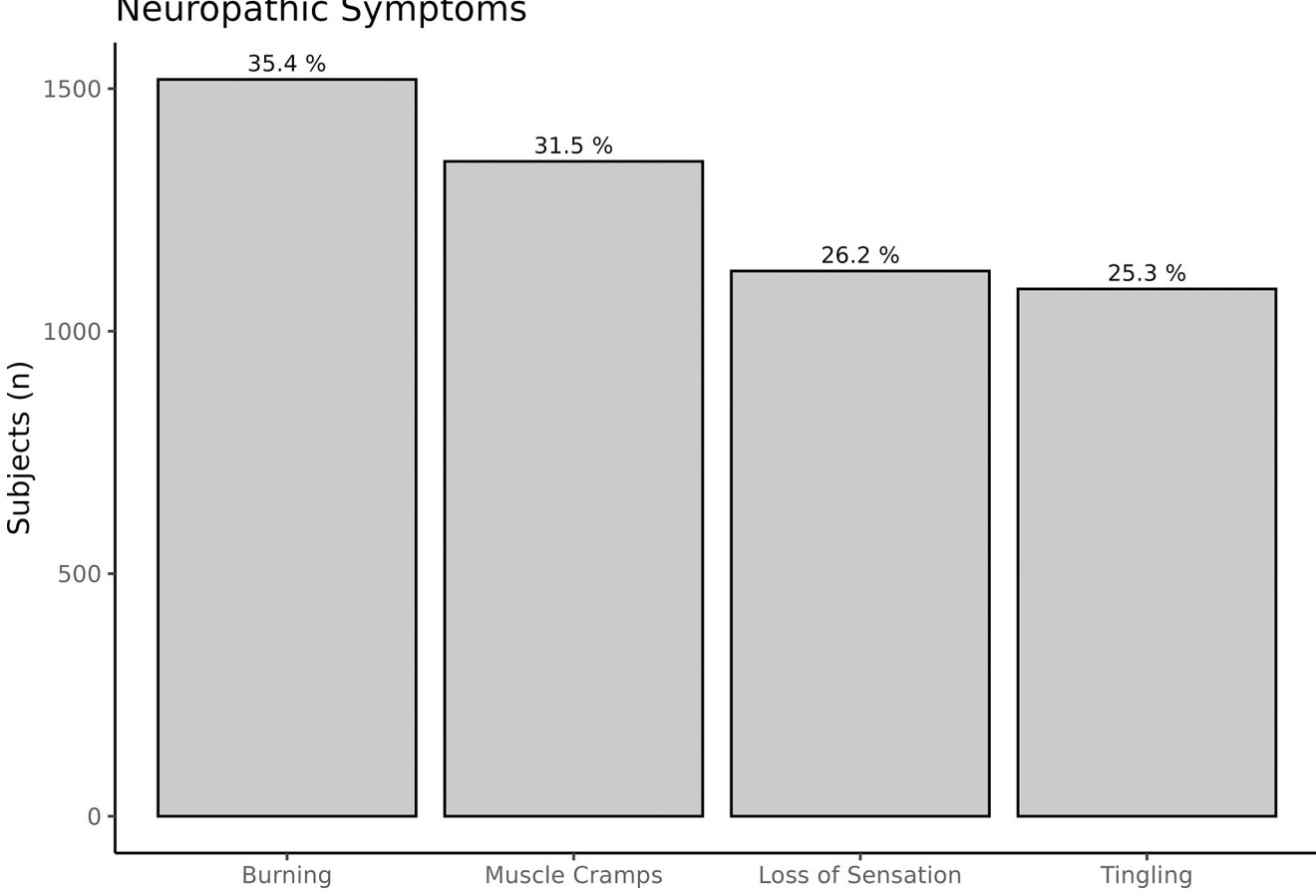

**Fig 2. The bar graph shows the percentage of subjects reporting different neuropathic symptoms.**

### Prevalence of foot deformities

Foot deformities, neuropathy, and high peak pressure on the foot are known risk factors for the onset of diabetic foot ulcers [14], which indicated a relative risk of 1.57 (95% CI: 1.22–2.02) for ulcer development within two years according to the study of Abott *et al.* [13].

In the present study, 79% of patients had at least one foot health-related issue. However, comparing these results with other studies is challenging. Most available data on diabetic neuropathy focus on ulcer prevalence, with limited information on other foot health complications. Additionally, variations in patient characteristics (age, risk factors, diabetes duration), study designs, and healthcare settings contribute to the wide range of prevalence rates in literature, making direct comparisons difficult.

In a high-risk diabetic cohort, Ledoux et al. reported a prevalence of 24% for bunions, 47% for hammer or claw toes, and 24% for hallux limitus [15]. These rates are higher than those found in our study, where 12% of patients had bunions and 10% had hammer toes. The lower prevalence in our population could be attributed to the inclusion of a broader group of patients, with a lower risk profile and a younger average age (51 vs. 62 years).

In a hemodialysis cohort study involving 232 patients with a mean age of 65 years, higher rates of foot complications were reported [16]. Similar to our study, dry skin was the most common issue. However, 64% of hemodialysis patients had dry skin, compared to 44% in our cohort. Additionally, 43% had thickened toenails, and other complications, such as fissured

**Table 2. Summary of the findings from logistic regression analysis, examining the relationship between various patient characteristics/health conditions and multiple foot health outcomes.**

| Predictors | Foot Health Complications n≥3 | | | Ingrown Nails | | | Toe deformities | | | Callus | | | Infection | | | Bunion | | | Dry Skin Foot | | | Maceration | | |
|---|---|---|---|---|---|---|---|---|---|---|---|---|---|---|---|---|---|---|---|---|---|---|---|---|
| | OR | CI | p | OR | CI | p | OR | CI | p | OR | CI | p | OR | CI | p | OR | CI | p | OR | CI | p | OR | CI | p |
| Male | 0.78 | 0.59–1.04 | 0.086 | 0.81 | 0.66–0.99 | 0.039 | | | | 0.91 | 0.84–1.00 | 0.052 | | | | | | | 1.23 | 1.17–1.29 | <0.001 | | | |
| HbA1c | 1.29 | 1.16–1.42 | <0.001 | | | | 1.09 | 1.00–1.18 | 0.049 | | | | | | | 0.90 | 0.83–0.98 | 0.012 | | | | | | |
| RBG | 1.00 | 0.99–1.00 | <0.001 | 1.00 | 1.00–1.00 | <0.001 | 1.00 | 1.00–1.00 | 0.010 | | | | 1.00 | 1.00–1.00 | 0.141 | | | | | | | | | |
| Smoking | 2.13 | 1.56–2.91 | <0.001 | 1.31 | 1.06–1.62 | 0.014 | | | | | | | 1.32 | 1.09–1.59 | 0.004 | 1.18 | 0.95–1.46 | 0.123 | | | | 1.49 | 1.23–1.80 | <0.001 |
| Alcohol IntakeYes | 1.57 | 1.14–2.15 | 0.006 | 1.34 | 1.09–1.66 | 0.006 | 1.44 | 1.15–1.79 | 0.001 | 1.58 | 1.25–1.98 | <0.001 | 1.22 | 1.01–1.47 | 0.039 | | | | | | | | | |
| Hypertension | 2.10 | 1.63–2.73 | <0.001 | | | | 1.26 | 1.03–1.56 | 0.028 | | | | 1.37 | 1.17–1.61 | <0.001 | | | | 1.45 | 1.28–1.65 | <0.001 | 1.24 | 1.04–1.49 | 0.019 |
| Dyslipidemia | 2.09 | 1.62–2.69 | <0.001 | 1.45 | 1.22–1.72 | <0.001 | 1.45 | 1.17–1.79 | 0.001 | 1.56 | 1.25–1.94 | <0.001 | 1.15 | 0.98–1.35 | 0.085 | 1.67 | 1.37–2.02 | <0.001 | 0.90 | 0.79–1.02 | 0.111 | | | |
| CAD | 3.48 | 2.42–5.04 | <0.001 | 1.49 | 1.16–1.89 | 0.001 | 1.42 | 1.04–1.90 | 0.022 | 1.77 | 1.32–2.36 | <0.001 | 1.53 | 1.21–1.92 | <0.001 | 1.98 | 1.52–2.57 | <0.001 | | | | 1.43 | 1.10–1.86 | 0.007 |
| Diabetes Nephropathy | 3.96 | 3.02–5.20 | <0.001 | 1.29 | 1.08–1.55 | 0.005 | 1.51 | 1.21–1.87 | <0.001 | 1.48 | 1.18–1.86 | 0.001 | 1.43 | 1.21–1.70 | <0.001 | 2.07 | 1.70–2.52 | <0.001 | 1.39 | 1.21–1.60 | <0.001 | 1.72 | 1.42–2.07 | <0.001 |
| Diabetic Retinopathy | 3.85 | 2.72–5.48 | <0.001 | 2.00 | 1.61–2.48 | <0.001 | 1.90 | 1.46–2.46 | <0.001 | 1.78 | 1.35–2.33 | <0.001 | 1.54 | 1.24–1.90 | <0.001 | 1.65 | 1.28–2.12 | <0.001 | | | | 1.44 | 1.14–1.82 | 0.002 |
| Neurological Condition | 1.76 | 1.15–2.69 | 0.009 | | | | 0.71 | 0.50–0.99 | 0.051 | 1.28 | 0.94–1.71 | 0.106 | | | | 1.50 | 1.15–1.94 | 0.002 | 1.35 | 1.12–1.63 | 0.002 | 1.63 | 1.27–2.07 | <0.001 |
| COVID History | 2.37 | 1.73–3.26 | <0.001 | | | | 1.36 | 1.05–1.73 | 0.017 | | | | 1.25 | 1.03–1.52 | 0.021 | 1.44 | 1.14–1.81 | 0.002 | 1.48 | 1.27–1.74 | <0.001 | 1.33 | 1.06–1.64 | 0.011 |
| Age | | | | 0.99 | 0.98–1.00 | 0.006 | 0.97 | 0.94–1.01 | 0.156 | | | | | | | 1.01 | 1.00–1.02 | 0.016 | | | | | | |
| Duration Of Diabetes | | | | 1.10 | 1.07–1.13 | <0.001 | | | | | | | | | | | | | | | | 0.95 | 0.92–0.98 | 0.001 |
| OAD | | | | | | | 1.66 | 1.18–2.41 | 0.005 | | | | 0.80 | 0.64–1.01 | 0.061 | 1.36 | 1.00–1.87 | 0.053 | | | | 0.62 | 0.49–0.79 | <0.001 |

**Table 3. Summary of the findings from logistic regression analysis, examining the associating between specific foot conditions and the presence of neuropathic symptoms in T2DM patients with DNP.**

| Predictors | Tingling | | | Burning | | | Loss of sensation | | | Muscle cramps | | |
|---|---|---|---|---|---|---|---|---|---|---|---|---|
| | OR | CI | p | OR | CI | p | OR | CI | p | OR | CI | p |
| Ingrown Nails | 1.64 | 1.37–1.95 | <**0.001** | 1.90 | 1.61–2.24 | <**0.001** | 1.26 | 1.05–1.51 | **0.013** | 1.69 | 1.42–2.01 | <**0.001** |
| Hammer Toe | 1.69 | 1.37–2.08 | <**0.001** | | | | 1.28 | 1.03–1.60 | **0.025** | 1.47 | 1.19–1.81 | <**0.001** |
| Callus | 1.55 | 1.23–1.94 | <**0.001** | 1.78 | 1.43–2.21 | <**0.001** | 1.56 | 1.23–1.96 | <**0.001** | 1.23 | 0.97–1.54 | 0.083 |
| Bunion | 1.44 | 1.17–1.76 | <**0.001** | 1.81 | 1.50–2.20 | <**0.001** | 1.39 | 1.13–1.71 | **0.002** | 1.33 | 1.09–1.63 | **0.006** |
| Dry Skin FootYes | 1.13 | 0.98–1.30 | 0.086 | 1.19 | 1.05–1.36 | **0.008** | 1.69 | 1.47–1.95 | <**0.001** | 2.01 | 1.76–2.30 | <**0.001** |
| Corn | 1.35 | 1.09–1.66 | **0.006** | 1.39 | 1.14–1.70 | **0.001** | 1.56 | 1.26–1.92 | <**0.001** | 2.14 | 1.75–2.62 | <**0.001** |
| Infection | | | | 1.61 | 1.38–1.88 | <**0.001** | 1.56 | 1.32–1.84 | <**0.001** | 1.17 | 0.99–1.38 | 0.065 |
| Maceration | | | | | | | 1.80 | 1.49–2.16 | <**0.001** | 1.81 | 1.51–2.18 | <**0.001** |

skin (34%) and a history of ulcers (24%), were more frequent. These higher findings likely reflect the higher risk profile of hemodialysis patients.

In the current study, 20% of patients had an infection, which suggests that many were under specialist care due to a prior diagnosis of neuropathy. In a study of 1666 patients who were enrolled in a program for preventing and treatment of diabetic foot complications, Lavery et al. reported that 9.1% of patients developed foot infections over two years [17]. Other studies have shown a lifetime risk of foot infections in diabetic patients ranging from 4% in primary care to an annual risk of 7% in tertiary care in the U.S. [18]. The high infection rate in the present study is concerning, as infections are the leading cause of diabetes-related hospitalizations and lower limb amputations.

## Prevalence of symptoms related to lower limb

A cross-sectional study by Katzberg et al. (2013), involving 144 patients with T2DM, found that the prevalence of muscle cramps was 75% [19]. This is notably higher than in the present study, where approximately one-third of patients experienced muscle cramps. This difference might be explained by age differences between the studies. Other studies have investigated the development and prevalence of symptoms over the course of the disease. It has been reported that around half of the patients with diabetic peripheral neuropathy experience asymmetric sensory changes [20], and that 30% to 40% of patients develops neuropathic pain during the disease course [21]. The present study found that 74% of patients had at least one symptom of diabetic neuropathy which could be due to selection bias or referral bias as all the centers were specialized in diabetes care and late referral practices in India when there is accumulation of diabetes complications including neuropathy.

## Prevalence of comorbidities

For the current study, the prevalence of diabetic retinopathy (13%) was low compared to the worldwide estimated prevalence of 22% among people with T2DM [22] and a previously reported 17% in an Indian (primarily southern) urban population [23]. In contrast, the prevalence of nephropathy (27%) aligns with other studies, which estimate that around 20% of people with type 2 diabetes worldwide develop diabetic nephropathy within 20 years of diabetes onset [24]. The possible underreporting of retinopathy compared to nephropathy, as observed in the present study, may be explained by differences in diagnostic approaches. Nephropathy was diagnosed using routine clinical examination, unlike retinopathy diagnosis that relied

primarily on medical history or the presence of symptoms, which may have contributed to under-reporting.

### Risk factors for foot abnormalities

Nevertheless, our recent data analysis has identified a strong association between diabetic retinopathy (OR: 3.85) and diabetic nephropathy (OR: 3.96) with the incidence of three or more foot health complications. This association supports the findings of previous research. A meta-analysis conducted by Li et al. examined the connection between diabetic retinopathy and the incidence of diabetic foot ulcers across 10,208 patients. Their findings indicate a significant increase in the risk of diabetic foot ulcers among patients with diabetic retinopathy, with an OR of 4.13 [25].

### Coronary artery disease and dyslipidemia

Also coronary artery disease (CAD) appeared to be strongly associated with having multiple foot health complications in the present study (OR: 3.48). This relationship was also found by, Meloni and colleagues who investigated the relationship between below-the-ankle (BTA) arterial disease and CAD in individuals with diabetic foot ulcers. Their study concluded that BTA arterial disease is an independent predictor of CAD, with an odds ratio of 1.9, highlighting the necessity of thorough vascular screening in diabetic patients, with or without foot ulcers [26,27].

Similarly, this study showed that dyslipidemia played a significant role in multiple foot health complications, with an OR of 2.1. A recent meta-analysis demonstrated a significant relationship between total cholesterol levels and the occurrence of diabetic foot ulcers, reporting an OR of 1.5 [28]. These findings highlight the critical role of dyslipidemia in the progression of diabetic foot problems through mechanisms like impaired wound healing and inflammation [29,30]. Additionally, it emphasizes the importance of managing lipid levels in diabetic patients to reduce the risk of these complications.

### COVID-19 and T2DM

Previous research has established a link between COVID-19 and T2DM, with a relative risk of 1.70 ([1.32–2.19] compared to non-COVID-19 patients) as reported by Zhang *et al.* [31]. Moreover, diabetic patients suffering from microvascular complications are more likely to experience severe outcomes from COVID-19 infection and may be at a higher risk of developing or exacerbating neuropathy and peripheral arterial disease [32,33]. The current analysis confirmed these earlier findings, showing that a history of COVID-19 infection could increase the likelihood of having toe deformities and/or three or more foot complications, with odds ratios of 1.36 ([1.05–1.73]) and 2.37 ([1.73–3.26]), respectively.

Key risk factors for DPN include increasing age, prolonged diabetes duration, and suboptimal glycemic control. However, also modifiable risk factors, including smoking, retinopathy, hypertension, obesity, hyperlipidemia, and microalbuminuria, are recognized as potential contributors to DPN risk [34].

Our research supports these findings, showing a significant correlation between HbA1c, alcohol use, hypertension, and dyslipidemia with several foot health complications, including the presence of three or more foot health complications. This indicates that adopting healthier lifestyles, maintaining tight glycemic control, and effectively managing hypertension and dyslipidemia can help preventing neuropathy-related deformities. The relatively low association of HbA1c with foot health issues, as compared to other factors, may be influenced by the fact that all participants were undergoing diabetes treatment at the time they were included for this

study. In addition, It is evident that there is a clear link between the duration of diabetes and foot health issues. The lack of correlations found in this study suggests that diabetes may often be diagnosed at a later stage and highlights the need for more rigorous diabetes screening strategies in India [35].

## Strengths and limitations

The present study should be seen in the context of its strengths and limitations. To our knowledge, it represents the largest cohort studied and most extensive examinations of foot health complications among T2DM patients with DNP in India. A structured approach was meticulously followed according to the study protocol for investigating and validating foot complications and symptoms. This methodical process ensured a consistent and systematic collection of data. Additionally, the data were gathered from multiple regions throughout India, ensuring a thorough representation of the Indian diabetic population.

However, the study also has its limitations. The evaluation of symptoms such as burning and tingling relied on patient self-reporting without quantifying the severity, which could lead to variance in symptom assessment. There was no differentiation among specific types of toe deformities due to the difficulty in clinical identification, which may have implications for the specificity of risk association with ulcer development. The omission of dorsalis pedis artery palpation in the study protocol suggests a potential oversight in the full vascular assessment of the diabetic foot. The diagnosis of neuropathy in this study was based solely on the physician's judgment. While this is common practice in India, it may have introduced some variability. A more standardized approach, such as using a validated questionnaire like the DN4 [36] for diagnosing neuropathy, would have been better to ensure consistency. Despite these limitations, the study reflects current clinical practice in India and provides valuable insights into the management of foot health among diabetic patients.

## Conclusions

The present study demonstrates a high prevalence of foot health conditions with most patients experiencing at least one DPN-related symptom. CAD, diabetic retinopathy and nephropathy were the predominant predictors associated with severe foot problems. In addition, a history of COVID-19 significantly increased the risk of foot complications, which illuminated the extensive impact of the pandemic on chronic disease management. Elevated HbA1c level, lifestyle behaviors such as smoking and alcohol consumption, co-morbid conditions like dyslipidemia and hypertension were linked to an increased risk of foot complications. These insights highlighted the necessity for comprehensive diabetes care that targets modifiable risks and closely monitors co-morbidities to prevent foot complications.

## Supporting information

**S1 File. Details of diabetic foot scanner tool.**
(DOCX)

**S1 Data. Final patient data sheet.**
(XLSX)

## Author Contributions

**Conceptualization:** Willem Jan Verberk.

**Data curation:** Ashu Rastogi, Manish Agarwal.

**Formal analysis:** Willem Jan Verberk.

**Writing – original draft:** Ashu Rastogi, Willem Jan Verberk.

**Writing – review & editing:** Ashu Rastogi, Venkatesan Ravindranath, Anupama Dubey, Dilip Gude, Manish Agarwal, Hiren Prajapati, Willem Jan Verberk.

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
