## [Decision Letter · Decision Letter 0]

21 Aug 2024

PONE-D-24-09121A Real-World Multicenter Cross-Sectional Observational Study To Assess The Clinical Profile Of PeRipheral Neuropathy In Patients with DiabetesPLOS ONE

Dear Dr. Rastogi,

Thank you for submitting your manuscript to PLOS ONE. After careful consideration, we feel that it has merit but does not fully meet PLOS ONE’s publication criteria as it currently stands. Therefore, we invite you to submit a revised version of the manuscript that addresses the points raised during the review process.

Overall this is a large and important contribution to the literature in relation to identifying DPN and foot problems in India.

The background should not be: Background: Increasing Type 2 Diabetes Mellitus (T2DM) in Asian-Indian populations highlights a gap in understanding diabetic peripheral neuropathy (DPN) and related foot deformities. Rather it should be: There are limited studies on the prevalence of diabetic peripheral neuropathy (DPN) and related foot deformities in patients with T2DM from India.In the methods you need to specify the study setting i.e. primary care, secondary care or tertiary care hospital?How many centers were involved and which part of India were they from?When assessing neuropathic symptoms was any particular validated questionnaire such as DN4 used?Details on the ‘Diabetic foot Scanner’ are required.How was ‘neuropathy’ diagnosed?The prevalence of diabetic retinopathy (12.9%) compared to nephropathy (27.2%) seems very low. How were these defined, from the medical notes or history?The prevalence of infection is very high (19.7%). Therefore were these patients at a specialist foot clinic?The discussion needs to be a discussion where the data from the current study has to be compared to previous papers from India or elsewhere. Currently it is simply a repeat of the results without any context. Thus, if 35.4% had burning symptoms, how does this compare to other studies?In the discussion you do not need to refer to the P value in previous studies.With regard to dyslipidemia, the discussion should focus on its relationship to DPN and DFU, not with CAD.Dry skin is not a good indicator of PAD, but represents sudomotor dysfunction. ==============================

We look forward to receiving your revised manuscript.

Kind regards,

Rayaz A. Malik, MBChB, PhD

Academic Editor

PLOS ONE

“This study was made possible with help of Eris Lifesciences Ltd.”

3. We note that your Data Availability Statement is currently as follows: [Data were uploaded as supporting material]

4. We note that there is identifying data in the Supporting Information file <file name>. Due to the inclusion of these potentially identifying data, we have removed this file from your file inventory. Prior to sharing human research participant data, authors should consult with an ethics committee to ensure data are shared in accordance with participant consent and all applicable local laws.

-Location data

Additional Editor Comments:

Overall this is a large and important contribution to the literature in relation to identifying DPN and foot problems in India.

1. The background should not be: Background: Increasing Type 2 Diabetes Mellitus (T2DM) in Asian-Indian populations highlights a gap in understanding diabetic peripheral neuropathy (DPN) and related foot deformities. Rather it should be: There are limited studies on the prevalence of diabetic peripheral neuropathy (DPN) and related foot deformities in patients with T2DM from India.

2. In the methods you need to specify the study setting i.e. primary care, secondary care or tertiary care hospital?

3. How many centers were involved and which part of India were they from?

4. When assessing neuropathic symptoms was any particular validated questionnaire such as DN4 used?

5. Details on the ‘Diabetic foot Scanner’ are required.

6. How was ‘neuropathy’ diagnosed?

7. The prevalence of diabetic retinopathy (12.9%) compared to nephropathy (27.2%) seems very low. How were these defined, from the medical notes or history?

8. The prevalence of infection is very high (19.7%). Therefore were these patients at a specialist foot clinic?

9. The discussion needs to be a discussion where the data from the current study has to be compared to previous papers from India or elsewhere. Currently it is simply a repeat of the results without any context. Thus, if 35.4% had burning symptoms, how does this compare to other studies?

10. In the discussion you do not need to refer to the P value in previous studies.

11. With regard to dyslipidemia, the discussion should focus on its relationship to DPN and DFU, not with CAD.

12. Dry skin is not a good indicator of PAD, but represents sudomotor dysfunction.

Reviewers' comments:

Reviewer's Responses to Questions

**Comments to the Author**

1. Is the manuscript technically sound, and do the data support the conclusions?

Reviewer #1: Partly

2. Has the statistical analysis been performed appropriately and rigorously? 

Reviewer #1: No

3. Have the authors made all data underlying the findings in their manuscript fully available?

Reviewer #1: Yes

4. Is the manuscript presented in an intelligible fashion and written in standard English?

Reviewer #1: No

5. Review Comments to the Author

Reviewer #1: In addition to limitations enumerated by authors, the data reliability is poor as there is no confirmation of presence or absence of complications by clinical or investigational methods. The bare foot walking, socio-economic status and working without covered foot wear, vegetarian/non vegetarian food intake and possibility of non-diabetic reasons for neuropathy-like symptoms are not reported. The study observations may not be equivalent if patients are from different regions reporting symptoms and assessed by different investigators. The details of 'Diabetic Foot Scanner' has not been provided and it's manufacturer information/data assessed by this instrument is not included.There is no details of vitamin B12 status of these patients.

6. PLOS authors have the option to publish the peer review history of their article (what does this mean?). If published, this will include your full peer review and any attached files.

Reviewer #1: No

---

## [Author Response · Author response to Decision Letter 0]

25 Sep 2024

¬Response to the reviewers

We sincerely thank the editor and reviewers for spending their valuable time to review our manuscript and for providing their comments, which helped to improve the quality of the manuscript. 

Point by point clarification/corrections to the concerns raised by the editor and reviewers are summarized. The changes in the revised manuscript are highlighted Red

Overall this is a large and important contribution to the literature in relation to identifying DPN and foot problems in India.

1. The background should not be: Background: Increasing Type 2 Diabetes Mellitus (T2DM) in Asian-Indian populations highlights a gap in understanding diabetic peripheral neuropathy (DPN) and related foot deformities. Rather it should be: There are limited studies on the prevalence of diabetic peripheral neuropathy (DPN) and related foot deformities in patients with T2DM from India.

We thank the reviewer for the suggestion and have changed accordingly. 

2. In the methods you need to specify the study setting i.e. primary care, secondary care or tertiary care hospital?

We have added the following sentence: “Data collection for the study was conducted between October 8, 2023, and November 20, 2023, with patients coming from primary, secondary, and tertiary care settings.”

3. How many centers were involved and which part of India were they from?

In the Material and Methods section we have added the following: 

Data collection for the study was conducted between October 8, 2023, and November 20, 2023, with patients coming from primary, secondary, and tertiary care settings across 543 centers in India, covering the following states: Jammu, Kashmir, Punjab, Rajasthan, Delhi, Bihar, Uttar Pradesh, Madhya Pradesh, Odisha, West Bengal, Assam, Andhra Pradesh, Tamil Nadu, Karnataka, Kerala, Maharashtra, and Gujarat.

4. When assessing neuropathic symptoms was any particular validated questionnaire such as DN4 used? The diagnosis of neuropathy in this study was based solely on the physician’s judgment. While this is common practice in India, it may have introduced some variability. A more standardized approach, such as using a validated questionnaire like the DN4 @bouhassira_comparison_2005 for diagnosing neuropathy, would have been better to ensure consistency.

5. Details on the ‘Diabetic foot Scanner’ are required.

We have provided a link to www.eris.co.in/diabetic-care , which includes all relevant information about the device.

6. How was ‘neuropathy’ diagnosed?

Neuropathy was diagnosed based on clinical judgement of practicing clinicians. We recognize that this is a limitation of the study and have mentioned this in the limitation section. “The diagnosis of neuropathy in this study was based solely on the physician’s judgment. While this is common practice in India, it may have introduced some variability. A more standardized approach, such as using a validated questionnaire like the DN4 @bouhassira_comparison_2005 for diagnosing neuropathy, would have been better to ensure consistency”

7. The prevalence of diabetic retinopathy (12.9%) compared to nephropathy (27.2%) seems very low. How were these defined, from the medical notes or history? In the discussion, we included a section on “Diabetic Peripheral Neuropathy and Comorbidities” to clarify that the results are based on patients' medical histories. Additionally, we explored potential reasons for the underreporting of retinopathy compared to nephropathy.

8. The prevalence of infection is very high (19.7%). Therefore were these patients at a specialist foot clinic?

Most of the centers are diabetes clinics, and patients were physically examined especially foot examination. Specialized diabetic foot clinic is not the usual norm in this part of the world and diabetes clinic provide foot care services as well which may not be called “specialized foot clinic” In the discussion section we added under the subheading: “Prevalence of Foot deformities” an explanation for the high number of patients with infection. 

9. The discussion needs to be a discussion where the data from the current study has to be compared to previous papers from India or elsewhere. Currently it is simply a repeat of the results without any context. Thus, if 35.4% had burning symptoms, how does this compare to other studies?

We thank the reviewer for this critical and valuable comment. We have checked for redundancy. Therefore we have rewritten major parts of the discussion section. We have mentioned the most important results included prevalence values and explained how they compare to previous studies. 

10. In the discussion you do not need to refer to the P value in previous studies.

We agree and have removed the p-values accordingly

11. With regard to dyslipidemia, the discussion should focus on its relationship to DPN and DFU, not with CAD.

We agree with the reviewer’s opinion and have removed the specific part of CAD relating to dyslipidemia. We only focused on the relationship of dyslipidemia with Foot health/DFU.

12. Dry skin is not a good indicator of PAD, but represents sudomotor dysfunction.

We agree and have edited the sentence: “This suggests it may be linked to reduced blood flow in PAD, although the association is not conclusive.” 

We hope that the revised manuscript may be considered suitable for acceptance.

Sincerely,

Ashu Rastogi

---

## [Editor Report · Decision Letter 1]

2 Oct 2024

A Real-World Multicenter Cross-Sectional Observational Study To Assess The Clinical Profile Of PeRipheral Neuropathy In Patients with Diabetes

PONE-D-24-09121R1

Dear Dr. Rastogi,

We’re pleased to inform you that your manuscript has been judged scientifically suitable for publication and will be formally accepted for publication once it meets all outstanding technical requirements.

Kind regards,

Rayaz A. Malik, MBChB, PhD

Academic Editor

PLOS ONE

Additional Editor Comments (optional):

Thank you for addressing the major concerns.
---

## [Editor Report · Acceptance letter]

14 Jan 2025

PONE-D-24-09121R1 

PLOS ONE

Dear Dr. Rastogi, 

I'm pleased to inform you that your manuscript has been deemed suitable for publication in PLOS ONE. Congratulations! Your manuscript is now being handed over to our production team.

Kind regards, 

on behalf of

Professor Rayaz A. Malik 

Academic Editor

PLOS ONE